# Advanced Mass Spectrometric Techniques for the Comprehensive Study of Synthesized Silicon-Based Silyl Organic Compounds: Identifying Fragmentation Pathways and Characterization

**DOI:** 10.3390/ma16093563

**Published:** 2023-05-06

**Authors:** Agnieszka Rogowska, Małgorzata Szultka-Młyńska, Basem Kanawati, Paweł Pomastowski, Adrian Arendowski, Adrian Gołębiowski, Phillipe Schmitt-Kopplin, Marta Fordymacka, Jarosław Sukiennik, Julia Krzywik, Bogusław Buszewski

**Affiliations:** 1Centre for Modern Interdisciplinary Technologies, Nicolaus Copernicus University in Torun, Wileńska 4, 87-100 Torun, Poland; aga4356@wp.pl (A.R.); pomastowski.pawel@gmail.com (P.P.); aarendowski@umk.pl (A.A.); adrian.golebiowski@doktorant.umk.pl (A.G.); bbusz@umk.pl (B.B.); 2Department of Environmental Chemistry and Bioanalytics, Faculty of Chemistry, Nicolaus Copernicus University in Torun, Gagarina 7, 87-100 Torun, Poland; 3Research Unit Analytical BioGeoChemistry, Helmholtz Center Munich—German Research Center for Environmental Health, 85764 Neuherberg, Germany; basem.kanawati@helmholtz-munich.de (B.K.); schmitt.kopplin@helmholtz-munich.de (P.S.-K.); 4TriMen Chemicals Sp. z o.o., Al. Piłsudskiego 141, 92-318 Lodz, Poland; mfordymacka@trimen.pl (M.F.); jsukiennik@trimen.pl (J.S.); jkrzywik@trimen.pl (J.K.)

**Keywords:** mass spectrometry, silicon-based silyl organic compounds, fragmentation pathway, organic synthesis

## Abstract

The primary objective of this study was to synthesize and characterize novel silicon-based silyl organic compounds in order to gain a deeper understanding of their potential applications and interactions with other compounds. Four new artificial silyl organic compounds were successfully synthesized: 1-O-(Trimethylsilyl)-2,3,4,6-tetra-O-acetyl-β-d-glucopyranose (compound **1**), 1-[(1,1-dimethylehtyl)diphenylsilyl]-1H-indole (compound **2**), O-tert-butyldiphenylsilyl-(3-hydroxypropyl)oleate (compound **3**), and 1-O-tert-Butyldiphenylsilyl-myo-inositol (compound **4**). To thoroughly characterize these synthesized compounds, a combination of advanced mass spectrometric techniques was employed, including nanoparticle-assisted laser desorption/ionization mass spectrometry (NALDI-MS), Fourier-transform ion cyclotron resonance mass spectrometry (FT-ICR-MS), and triple quadrupole electrospray tandem mass spectrometry (QqQ ESI-MS/MS). These analytical methods enabled the accurate identification and characterization of the synthesized silyl organic compounds, providing valuable insights into their properties and potential applications. Furthermore, the electrospray ionization–Fourier transform ion cyclotron resonance–tandem mass spectrometry (ESI-FT-ICR-MS/MS) technique facilitated the proposal of fragmentation pathways for the ionized silyl organic compounds, contributing to a more comprehensive understanding of their behavior during mass spectrometric analysis. These findings suggest that mass spectrometric techniques offer a highly effective means of investigating and characterizing naturally occurring silicon-based silyl organic compounds, with potential implications for advancing research in various fields and applications in different industries.

## 1. Introduction

For centuries, therapeutic properties have been attributed to various natural sources [1]. The key to understanding these therapeutic qualities lies in the knowledge of their chemical composition [2]. With advancements in science, especially in the field of metabolomics, it has become possible to identify specific compounds responsible for unique health-promoting properties. A wide array of metabolites can be produced, which are utilized in industries, such as pharmaceuticals, biotechnology, and medicine [3]. These compounds are primarily valued for their antioxidant, antimicrobial, antiviral, and anti-inflammatory properties, along with low cytotoxicity [1,4,5].

Furthermore, the ongoing evolution of genes enables the synthesis of new metabolites with varying properties [3]. Although primary metabolites exhibit similarities across all living cells, the production of secondary metabolites is influenced by the specific conditions under which they grow [2]. Secondary metabolites, which are synthesized by cells via metabolic pathways originating from primary metabolic pathways [2], seem to hold particular therapeutic potential. While not essential for an organism’s survival, these compounds serve to protect against harmful external factors and play a crucial role in adaptation to both abiotic and biotic stress conditions [6]. Consequently, optimizing specific growth conditions can stimulate the synthesis of known and new metabolites with potential therapeutic applications.

Silicon, which is the second most abundant element in the soil, after oxygen, has a beneficial effect on the growth of plants, especially those exposed to unfavorable environmental conditions [7,8,9]. This element is resorbed by plants from the soil in a form of silicic acid Si(OH)_4_, and its uptake efficiency depends on the plant species [7,9,10]. Therefore, the Si content in dry weight can vary from 0.1 to 10% [9,11]. It was proved that silicon reduces the negative effects of biotic (e.g., resistance to viruses, as well as bacterial and fungal pathogens) and abiotic stress (such as exposure to heavy metals, cold, heat, salt, water, or UV-B) [7,8,9,12,13]. Initially, it was thought that the protective properties of silicon were related to the formation of a physical barrier that strengthened the cell wall. However, more detailed studies indicated a much more complex mechanism of the silicon action on plants, which may involve interactions with the interior of the cell and changes in metabolism [7,13]. It was also shown that Si was able to modulate the secondary metabolism of plants, increasing their stress tolerance [9]. Although some influence of silicon on the gene expression and plant was shown, the exact mechanisms contributing to the plant growth during the presence of this element are still ambiguous [8].

While numerous studies have explored the impact of silicon on plant growth and secondary metabolite production, there is no direct evidence for the formation of silicon derivatives of organic compounds in plants. The application of nuclear magnetic resonance (NMR) and Fourier transform infrared (FT-IR) spectroscopy, as employed by Cabrera et al. [14], confirmed that lignin can interact with silicon through hydrogen bonds formed between lignin’s aliphatic carboxyl or hydroxyl groups and silanols. However, the creation of C-O-Si bonds and silica condensation in lignin do not occur. Additionally, Inanaga et al. [15] suggested the potential existence of some organic compounds combined with silicon in rice cell walls using ultraviolet (UV) and infrared (IR) methods. As the mechanisms of interactions between organic compounds present in plants and silicon remain unknown, it is crucial to explore methods that enable the identification of such silica-based organic compounds. Mass spectrometry techniques present promising and continually evolving analytical approaches for analyzing silicon derivatives. In the initial stage of this research, it is essential to develop a workflow using spectroscopic and spectrometric methods for their potential identification.

Consequently, this study aimed to employ various mass spectrometry techniques to characterize artificially synthesized silicon-based silyl organic compounds. To this end, four proposed organic derivatives were synthesized: 1-O-(Trimethylsilyl)-2,3,4,6-tetra-O-acetyl-β-d-glucopyranose, 1-[(1,1-dimethylehtyl)diphenylsilyl]-1H-indole, O-tert-butyldiphenylsilyl-(3-hydroxypropyl)oleate, and 1-O-tert-Butyldiphenylsilyl-myo-inositol.

The synthesized compounds were analyzed using silver nanoparticle-assisted laser desorption/ionization mass spectrometry (NALDI-MS), Fourier-transform ion cyclotron resonance mass spectrometry (FT-ICR-MS), and triple-quadrupole tandem electrospray ionization mass spectrometry (QqQ ESI-MS/MS) techniques.

## 2. Materials and Methods

### 2.1. Chemicals and Reagents

All solvents, substrates, and reagents for synthesis of silicon-based silyl organic compounds were obtained from TriMen Chemicals (Lodz, Poland) and were used without further purification. All chemicals used for mass spectrometry analysis, e.g., chromatographic grade pure acetonitrile, water, and formic acid, were purchased from Merck (Darmstadt, Germany).

### 2.2. Synthesis of 1-O-(Trimethylsilyl)-2,3,4,6-tetra-O-acetyl-β-d-glucopyranose (Compound ***1***)

The synthesis of compound **1** (Si-pyranose) was carried out on the basis of Vedachalam et al.’s [16] methodology. Methylamine in THF (2 M, 1.5 mL) was added dropwise to a suspension of pentaacetyl glucose (1 g, 2.56 mmol) in dry THF (10 mL) at 0 °C. The reaction mixture was stirred for 3 h.

The solvent evaporated, and the residue was dissolved in dichloromethane (100 mL) and washed three times with water and with brine. The solution was dried with MgSO_4_ and evaporated to give tetraacetyl glucose, which is immediately used for the next step (0.79 g, 95%).

Crude tetraacetyl glucose obtained as above (0.79 g, 2.27 mmol) was dissolved in dichloromethane (20 mL), containing triethylamine (0.38 mL, 2.7 mmol); next, chlorotrimethylsilane (0.313 mL, 2.5 mmol) was added at room temperature. After being stirred for 2 h, the mixture was washed with water, 5% sodium bicarbonate, brine, and was dried with MgSO_4_. After the removal of the solvent, the residue was purified by flash chromatography, using 96:4 hexane-EtOAc solvent system to make the desired product crystalline solid (0.44 g, 86% yield, 99% purity) as a single anomer. The thin layer chromatography (TLC) visualization was performed with the 5% ethanolic solution of anisaldehyde.

### 2.3. Synthesis of 1-[(1,1-Dimethylehtyl)diphenylsilyl]-1H-indole (Compound ***2***)

The synthesis of compound **2** (Si-indole) was carried out on the basis of Medina-Mercado et al.’s [17] methodology. To a solution of indole (1 g, 8.5 mmol) in anhydrous DMF (10 mL), NaH (60%, 0.512 g, 12.8 mmol) was added at 0 °C under argon protecting gas. The reaction mixture was warmed to room temperature (RT) over 1.5 h. After that, tert-butyldiphenylsilyl chloride (3.39 mL, 13 mmol) was added dropwise, and the mixture was stirred at RT overnight. Subsequently, the reaction mixture was washed and diluted with ethyl acetate (100 mL), washed with water, 1M aqueous HCl, 5% sodium bicarbonate, and brine. The combined organic extracts were dried over MgSO_4_ and were concentrated under the vacuum. The obtained residue was further purified by column chromatography on the silica gel, using 90:10 hexane-EtOAc as eluent, to give a beige waxy solid (300 mg, 80% yield, 99% purity).

### 2.4. Synthesis of O-Tert-butyldiphenylsilyl-(3-hydroxypropyl)oleate (Compound ***3***)

To a solution of sodium hydride, NaH (60%, 6 mmol, 240 mg) in tetrahydrofurane THF (10 mL), cooled in an ice bath, along with 1,3-propanediol (6 mmol, 456 mg) diluted with THF (5 mL), were added dropwise. After 30 min, tert-butyldiphenylchlorosilane TBDPS-Cl (6 mmol, 1.6 mL) was diluted with THF (5 mL) and was added to the mixture. Subsequently, the ice bath was removed, and the reaction was continued at RT. The progress was monitored by liquid chromatography coupled to a mass spectrometer (LC-MS). Subsequently, the reaction mixture was quenched with H_2_O and diluted with Et_2_O (50 mL) and EtOAc (50 mL). Then, the organic layer was washed with H_2_O, brine and dried over MgSO_4_. The solvent evaporated, and the residue was used in the next step, without purification.

Oleoyl chloride (85% purity, 6 mmol, 2.1 g) diluted with THF (5 mL) was added dropwise to a solution of the above crude TBDPS-protected alcohol (1.88 g 6 mmol), 4-dimethylaminopyridine, DMAP (0.1 equiv. 73 mg), triethylamine, and TEA (12 mmol, 1.67 mL) in THF (10 mL). The reaction progress was monitored by LC-MS. Then, the reaction mixture was diluted with EtOAc (100 mL) and washed once with H_2_O, twice with 2 M HCl, twice with 5% NaHCO_3_, once with brine, and dried over MgSO_4_. The residue was purified using the column flash chromatography (silica gel; EtOAc/hexanes) to give the pure product (99% purity) in the form of colorless oil with an overall yield of 1.9 g, 95% after the two above-mentioned steps.

### 2.5. Synthesis of 1-O-Tert-Butyldiphenylsilyl-myo-inositol (Compound ***4***)

The synthesis of 1-O-tert-Butyldiphenylsilyl-myo-inositol (Si-inositol) was carried out on the basis of Watanabe et al.’s [18] methodology. Pyridine (6 mL), DMAP (122 mg, 1 mmol), and TBDPS-Cl (2.3 mL, 8.88 mmol) were added to a solution of myo-inositol (2 g, 11.1 mmol) in DMSO (10 mL), and the mixture was left overnight at RT. The mixture was diluted with ethyl acetate (100 mL) and washed three times with water. It was brined, dried (MgSO_4_), and it evaporated. The residue was purified by flash chromatography using the dichloromethane/methanol gradient system, starting with 1% methanol to give thick colorless oil that was freeze-dried from the aqueous suspension to give a waxy solid—320 mg, 88% yield, over 95% purity.

### 2.6. NMR Analysis

NMR spectra were recorded on Bruker Avance 700 magnetic resonance spectrometers (Bruker, Billerica, MA, USA). ^1^H^-^NMR spectra are reported in chemical shifts downfield from TMS tetramethylsilane (TMS) using the respective residual solvent peak as the internal standard (CDCl_3_ δ 7.26 ppm, (CD_3_)_2_SO δ 2.50 ppm) (Appendix A). ^13^C^-^NMR spectra are reported in chemical shifts downfield from TMS using the respective residual solvent peak as the internal standard (CDCl_3_ δ 77.2 ppm, (CD_3_)_2_SO δ 39.5 ppm).

### 2.7. Compound Analysis Using Mass Spectrometry Techniques

The compounds were suspended in LC-MS grade acetonitrile to obtain a final concentration of 10 ppm, and the characteristics of silicon-based silyl organic compounds synthesized in accordance to Section 2.3, Section 2.4, Section 2.5 and Section 2.6 have been conducted.

#### 2.7.1. NALDI/SALDI MS Analysis on Silver Nanoparticles

A surface-assisted laser desorption/ionization (SALDI) target covered with silver nanoparticles was prepared in a similar manner to that described by Nizioł et al. [19]. A stainless steel plate of 75 mm × 25 mm × 0.8 mm size (H17 grade, locally made) was inserted into a large, glass Petri dish containing 50 mL of tetrahydrofuran (EMPLURA, Supelco, Merck) with dissolved 10 mg of silver trifluoroacetate (98%, Sigma–Aldrich, Schnelldorf, Germany) and 35 mg of 2,5-dihydroxybenzoic acid (99%, Bruker Daltonics GmbH, Bremen, Germany). After 36 h of reaction, the silver target plate was washed several times with tetrahydrofuran, wiped with a cotton wool ball, and washed three times with hot isopropanol (LC-MS, Supelco, Merck, Darmstadt, Germany) and acetonitrile (LC-MS, Fluka, Steinheim, Germany). The target was used with Bruker MTP Slide–Adapter II. The volumes of 0.5 mL of samples were placed directly on the target plate, air dried, and inserted into the MS apparatus for measurements.

A SALDI MS experiment was performed using the Bruker ultraFlextreme Time-of-Flight mass spectrometer (Bruker Daltonics GmbH & Co., Bremen, Germany), equipped with a SmartBeam II laser (Wavelength: 355 nm and frequency 2 kHz) in the positive-ion reflectron mode. The measurement range was *m*/*z* 80–2000, while the suppression was turned on for *m*/*z* lower than 79. The number of laser shots was 2000 (4 × 500 shots) for each sample spot. The first accelerating voltage was held at 25.08 kV, and the second ion source voltage was held at 22.43 kV. Reflector voltages are 26.64 and 13.54 kV. The value of the detector gain for the reflector was 5×. The value of the global attenuator offset accounted for 42%. Mass calibration was performed with FlexAnalysis 3.3 (Bruker Daltonics GmbH), using the cubic enhanced model and internal standards (silver ions and clusters from Ag^+^ to Ag_4_^+^). The theoretical *m*/*z* values of the analyzed compounds were confirmed by using the web page chemcalc.org [20].

#### 2.7.2. FT-ICR-MS Analysis

The FT-ICR-MS analysis was conducted using a Solarix Fourier transform ion cyclotron resonance mass spectrometer (FT-ICR MS) (Bruker Daltonics GmbH, Bremen, Germany) coupled to a 12 Tesla superconducting magnet (Magnex Scientific Inc., Yarnton, GB). The sample direct infusion was conducted by an APOLO II electrospray (ESI) source (Bruker Daltonics GmbH, Bremen, Germany) in the positive ionization mode. The sample was injected manually (direct infusion mode), and the capillary was flushed with ultrapure acetonitrile before and after each injection. The calibration of the obtained spectra was performed externally by applying the arginine solution (10 ppm in acetonitrile). For each analysis, 50 scans were collected in the *m*/*z* range (123–1000) amu. For better sensitivity, the ions were accumulated in the collision cell (hexapole) for 300 ms prior to their entrance to the ICR cell. The electrospray ionization voltage of the capillary was 3600 V, and the applied spray shield voltage was −500 V. The dry nitrogen gas flow rate and the temperature of the drying nitrogen gas were set to 4.0 L/min and 200 °C, respectively. The data were analyzed using the Data Analysis Version 4.2 (Bruker Daltonics GmbH, Bremen, Germany).

#### 2.7.3. QqQ ESI-MS/MS Analysis

The determination and identification of the studied silicon-based organic compounds were carried out using a triple quadrupole mass spectrometer (LC-MS/MS 8050 Shimadzu (Kyoto, Japan)) equipped with the LabSolution software (version 5.8) for data collection and instrumental control. The ESI-MS/MS spectrometer was coupled to an Ultra high pressure liquid chromatography (UHPLC) system (LC-30AD binary solvent delivery system, SIL-30AC autosampler, and CTO-20AC thermostat) (Kyoto, Japan). Electrospray ionization (ESI) was applied in both the positive and negative ion modes. The optimization of different MS parameters on the selectivity and MS response (multiple reaction monitoring, MRM peak areas) for the studied compounds was carried out without a chromatographic column. The optimal parameters were as follows: the interface temperature: 275 °C, the desolvation line (DL) temperature: 250 °C, the nebulizing gas flow rate: 4 L/min, the heating gas flow rate: 8 L/min, and the temperature of the drying gas: 360 °C. The studied compounds were monitored in the scheduled multiple reaction monitoring (MRM) mode. The total dwell time was 0.7 s. Chromatographic XDB-C18 column (2.1 mm × 50 mm, 1.8 µm, Agilent Technologies, Waldbronn, Germany) was maintained at 25 ± 0.5 °C. The injected sample volume was 1 µL, while the mobile phase was composed of acetonitrile (ACN) + 0.1% HCOOH (35:65, *v*/*v*) dosed at a flow rate of 0.3 mL/min. The addition of 0.1% formic acid to the acetonitrile/water mobile phase could improve peak shapes and increase the MS detection sensitivity. The data acquisition and processing were carried out using the LabSolution Workstation software (version 5.8). The data were further processed using Microsoft Excel.

## 3. Results and Discussion

### 3.1. Characteristics of the Obtained Organic Compounds Using NMR Approach

The synthesis pathway of 1-O-(Trimethylsilyl)-2,3,4,6-tetra-O-acetyl-β-d-glucopyranose (Si-pyranose) is shown in Figure 1A.

^1^H-NMR (700 MHz, CDCl_3_) δ 5.21 (t, *J* = 9.6 Hz, 1H), 5.12–5.03 (m, 1H), 4.96–4.89 (m, 1H), 4.77 (d, *J* = 7.7 Hz, 1H), 4.23 (dd, *J* = 12.2, 5.6 Hz, 1H), 4.15 (dd, *J* = 12.1, 2.5 Hz, 1H), 3.73 (ddd, *J* = 10.0, 5.6, 2.5 Hz, 1H), 2.09 (s, 3H), 2.07–2.05 (m, 3H), 2.05–2.04 (m, 3H), 2.02 (d, *J* = 2.8 Hz, 3H), 0.20–0.15 (m, 9H) (Appendix A).

^13^C-NMR (176 MHz, CDCl_3_) δ 170.57, 170.28, 169.41, 169.27, 95.56, 73.24, 72.77, 71.90, 68.71, 62.22, 20.66, 20.65, 20.60, 20.57, −0.04 (Figure 1).

The synthesis pathway of compound **2**: 1-[(1,1-dimethylehtyl)diphenylsilyl]-1H-indole (Si-indole) is shown in Figure 1B.

^1^H-NMR (700 MHz, CDCl_3_) δ 7.73–7.69 (m, 1H), 7.68–7.64 (m, 4H), 7.53–7.47 (m, 3H), 7.45–7.40 (m, 4H), 7.12 (ddd, J = 7.9, 6.9, 1.0 Hz, 1H), 6.91 (dd, J = 7.0, 1.3 Hz, 1H), 6.86 (dd, J = 8.4, 0.8 Hz, 1H), 6.77 (dd, J = 3.2, 0.8 Hz, 1H), 1.36–1.24 (m, 9H) (Appendix A).

^13^C-NMR (176 MHz, CDCl_3_) δ 141.02 (s), 135.87 (s), 132.35 (s), 131.94 (s), 131.49 (s), 130.27 (s), 128.18 (s), 121.21 (s), 120.42 (s), 120.10 (s), 115.56 (s), 105.53 (s), 28.38 (s), 20.01 (s) (Figure 2).

The synthesis pathway of compound **3**: O-tert-butyldiphenylsilyl-(3-hydroxypropyl)oleate (Si-oleate) is shown in Figure 1C.

^1^H-NMR (700 MHz, CDCl_3_) δ 7.72–7.69 (m, 4H), 7.46 (ddd, J = 8.9, 2.9, 1.5 Hz, 2H), 7.42 (ddd, J = 8.0, 4.4, 1.0 Hz, 4H), 5.45–5.34 (m, 2H), 4.27 (t, J = 6.4 Hz, 2H), 3.78 (t, J = 6.0 Hz, 2H), 2.32–2.25 (m, 2H), 2.08–2.03 (m, 4H), 1.91 (p, J = 6.2 Hz, 2H), 1.67–1.61 (m, 2H), 1.39–1.28 (m, 20H), 1.09 (s, 9H), 0.92 (t, J = 7.1 Hz, 3H) ppm (Appendix A).

^13^C-NMR (176 MHz, CDCl_3_) δ 173.8, 135.6, 133.7, 130.0, 129.8, 129.6, 127.7, 61.2, 60.3, 34.3, 31.9, 31.7, 29.8, 29.7, 29.5, 29.3, 29.3, 29.2, 29.2, 29.1, 27.3, 27.2, 26.9, 25.0, 22.7, 19.2, 14.1 ppm (Figure 3).

The synthesis pathway of compound **4**: 1-O-tert-Butyldiphenylsilyl-myo-inositol (Si-inositol) is shown in Figure 1D.

^1^H-NMR (700 MHz, DMSO) δ 7.80–7.68 (m, 4H), 7.46–7.33 (m, 6H), 4.60 (d, *J* = 4.3 Hz, 1H), 4.58 (d, *J* = 4.8 Hz, 1H), 4.45 (t, *J* = 7.6 Hz, 1H), 4.42 (d, *J* = 5.1 Hz, 1H), 4.39 (d, *J* = 3.1 Hz, 1H), 3.64 (td, *J* = 9.3, 4.7 Hz, 1H), 3.52 (d, *J* = 2.6 Hz, 1H), 2.85–2.76 (m, 2H), 1.02 (s, 9H) (Appendix A).

^13^C-NMR (101 MHz, DMSO) δ 136.65, 136.28, 136.03, 134.95, 134.10, 129.92, 127.95, 127.88, 127.52, 75.81, 74.79, 73.48, 73.12, 72.90, 72.21, 40.53, 40.32, 40.12, 39.91, 39.70, 39.49, 39.28, 27.63, 27.38, 19.68 (Figure 4).

The applications of the discussed compounds are notably diverse, primarily serving as intermediates or precursors in the synthesis of more complex molecules or materials. Additionally, selected synthesis procedures for related compounds have been included in the Appendix A. The compounds and their derivatives, mentioned in Appendix A, can be useful as reference materials or internal standards in mass spectrometry analysis. They can assist in the precise identification and quantification of target analytes. Using these compounds as internal standards can improve the accuracy and consistency of measurements by accounting for any variations that may arise during sample preparation, instrument performance, or matrix effects. Consequently, the use of these compounds in mass spectrometry analysis has contributed to the progress of scientific research in various fields, such as pharmaceuticals, environmental monitoring, and biochemistry. Some potential applications for each compound are as follows: 1-O-(Trimethylsilyl)-2,3,4,6-tetra-O-acetyl-β-d-glucopyranose is employed as a precursor in crafting glycosidic linkages for oligosaccharides and glycoconjugates [21]. It serves as a starting point for producing glycosyl donors in glycosylation reactions [22]. Additionally, it functions as an intermediate in synthesizing modified carbohydrates for drug discovery [23]. 1-[(1,1-dimethylehtyl)diphenylsilyl]-1H-indole can be utilized as a precursor to prepare indole-based heterocyclic compounds [24]. Regarding O-tert-butyldiphenylsilyl-(3-hydroxypropyl)oleate, it can function as an intermediate in creating oleate-based surfactants and detergents [25]. Furthermore, it can act as a precursor for preparing oleate-based functionalized lipids [26]. Lastly, 1-O-tert-Butyldiphenylsilyl-myo-inositol can be employed as an intermediate in synthesizing inositol-containing natural products and pharmaceuticals [27].

### 3.2. Characteristics of the Obtained Organic Compounds Using Mass Spectrometry Techniques

In order to accurately characterize the compounds synthetized as described in Section 2.3, Section 2.4, Section 2.5 and Section 2.6, the three mass spectrometric techniques were applied.

Table 1 summarizes the obtained *m*/*z* values of the parent ions with different adducts registered on the NALDI-MS, FT-ICR MS, and QqQ ESI-MS/MS mass spectra for each of the investigated silyl organic compounds (**1**–**4**).

In the first stage of the research, an attempt was made to analyze the obtained compounds using the matrix-assisted laser desorption/ionization (MALDI) mass spectrometry. However, this method turned out to be ineffective and did not enable obtaining the required MS signals, which characterize the analyzed silyl organic compounds (Appendix A). For this reason, a modern approach based on the silver nanoparticle-assisted laser desorption/ionization mass spectrometry (NALDI-MS) was used. This method allows performing the analysis without the matrix application. The NALDI-MS spectra were recorded in the positive ion detection mode. First, the signals for the plate coated with silver nanoparticles were recorded (Appendix A). The signals at *m*/*z* of 106.905, 215.811, and 322.713 were derived from Ag^+^, Ag_2_^+^, and Ag_3_^+^ ions, respectively [28]. For Ag^+^, two isotopes could be easily distinguished at nominal *m*/*z* 107 and 109. With the increasing number of silver atoms in the cluster ions (Ag_2_^+^ and Ag_3_^+^), more isotopes can be detected.

In addition, Figure 5 and Appendix A summarize the NALDI-MS spectra of compounds (**1**–**4**). On each of the spectra, the dominant signals from the silver nanoparticles, which covered the SALDI target, could be observed along with the less intense signals specific to individual compounds (**1**–**4**). Signals from the analyzed compounds emerge in the form of sodium and silver adducts (Table 1). The identification of signals from silver adducts was carried out on the basis of specific signal envelopes. It was possible to observe signals from a given compound in combination with both silver isotopes ^107^Ag and ^109^Ag. Thus, it can be concluded that, in contrast to the traditional MALDI technique, the use of NALDI functions significantly for the identification and characterization of the target compounds (**1**–**4**). Moreover, an attempt was made to fragment the obtained precursor ions using the LIFT mode. However, the use of the NALDI LIFT-TOF/TOF MS technique turned out to be unsuitable for the study of the fragmentation paths of the described organosilicon compounds.

Fourier-transform ion cyclotron resonance mass spectrometry (FT-ICR-MS), which is a high-resolution technique, was also implemented to determine highly accurate masses of the investigated silyl organic compounds (**1**–**4**). All of the MS spectra were acquired in the positive ion mode of electrospray. Figure 6 presents the obtained ESI+FT-ICR MS spectra of Si-pyranose, Si-indole, Si-oleate, and Si-inositol. The use of this technique, as opposed to the NALDI technique, did indeed make it possible to successfully identify all of the tested compounds. For Si-pyranose, Si-oleate, and Si-inositol, the presence of sodium adducts was observed in their corresponding FT-ICR-MS spectra (Figure 6A,C,D) at *m*/*z* 443, 601, and 441, respectively. In the case of compound **2**: Si-indole, a signal from the protonated ion at *m*/*z* 356 was observed. This is because of the existence of a nitrogen atom in the Si-indol compound, which enables the proton attachment to that N-atom due to the high gas phase proton affinity of nitrogen atoms in general. For compound **3**: Si-oleate, the presence of signals from both sodiated (*m*/*z* 601) and protonated (*m*/*z* 579) ions was confirmed, with the dominance of the sodium adduct ion. Moreover, in the case of this compound, partial fragmentation in the ion source occurred. This is proven by the presence of the fragment ion at *m*/*z* 501.

Finally, the compounds were also analyzed using the QqQ ESI-MS/MS system, and the obtained parent ions were fragmented to obtain the product ions to allow the trace of the fragmentation pathway. For compound **1**: Si-pyranose, the ionization was performed in the positive ion mode, and the identified parent ion was at *m*/*z* 443 for [M+Na]^+^. Figure 7 shows the MS/MS spectrum of compound **1**: Si-pyranose, which shows [M+Na]^+^ *m*/*z* 443, along with its fragment ions (Figure 7). The most intensive fragment ion signal in the FT-ICR-MS/MS spectrum was observed at *m*/*z* 383. This ion can be assigned to 1-O-(Trimethylsilyl)-2,3,4,6-tetra-O-acetyl-β-d-glucopyranose after the disconnection of one –OAc group and the elimination of acetic acid. In addition, the signal with low intensity at *m/z* 241 can be attributed to the sodiated [C_9_H_14_O_6_ + Na]^+^ ion.

The positive ion mode of FT-ICR was also applied in order to acquire the MS/MS spectrum of Compound **2**: Si-indole. The identified parent ion appeared at *m*/*z* 356 and represents protonated Si-indole [M+H]^+^. Figure 8 shows the MS/MS spectrum of [M+H]^+^ *m*/*z* 356 of Si-indole and its diverse precursor ion fragmentation. Ion [M+H]^+^ *m*/*z* 356 is generated as a result of Si–indole ionization by the proton attachment to the N-atom in the indole moiety. Once formed and accelerated, this ion can detach either tert-Butyl group to form the protonated radical ion *m*/*z* 299, or it can detach one of the two connected benzyl groups to form the product ion *m*/*z* 278, which presents a tri-coordinated silicon ion. Tri-coordinated silicon ions are very well known in gas phase mass spectrometric studies [29,30]. Moreover, another possible product ion can be formed as a result of the MS/MS fragmentation, when the indol moiety is released to form ion *m*/*z* 239. A gas phase collision-induced rearrangement of this ion can produce the fragment ion *m*/*z* 197. Ion *m*/*z* 135 may represent a tri-coordinated benzyl–dimethyl silicon ion. Further isolation of ion *m*/*z* 135 was not possible, so that further structural elucidation of that fragment ion was not feasible.

The QqQ ESI-MS/MS spectrum of Si-oleate in the positive ion mode reveals the parent ion at *m*/*z* 579 in a protonated form [M+H]^+^. The MS/MS spectrum of this precursor ion [M+H]^+^ *m*/*z* 579 shows extensive fragmentation and is summarized in Figure 9. Ions with representative signals at *m*/*z* 501, 443, and 265 could also be detected in FT-ICR-MS. The signal at *m*/*z* 501 can be attributed to the product ion obtained after the detachment of one phenyl ring from the parent ion. The ion at *m*/*z* 443 originates from ion *m*/*z* 501 as a result of the elimination of the tert-butyl group and the formation of an additional double bond in the hydrophobic chain of the oleate moiety. The product ion *m*/*z* 323 can be formed as a result of detachment of a phenylsilanol group from ion *m*/*z* 443. In turn, the signal at *m/z* 265 represents a product ion with the sum formula [C_18_H_32_O + H]^+^, which may be formed after detaching of propanol moiety from ion *m*/*z* 323. Further fragmentation of ion *m*/*z* 265 is possible by releasing the aldehydic group to produce ion *m*/*z* 237.

In the case of Compound **4**: Si-inositol, the MS/MS spectra were acquired in the negative ion mode using the QqQ-ESI-MS/MS system. Figure 10 shows the MS/MS spectrum of [M-H]^−^ *m*/*z* 417 ion fragmentation, with the proposed fragmentation pattern of Si-inositol. Two main signals from the product ion were revealed. The most intensive signal at *m*/*z* 255 can be formed due to the detachment of the inositol group from the parent compound. In turn, the signal at *m*/*z* 161 corresponds to the detached inositol moiety after conversion of one hydroxyl group to a carbonyl group.

## 4. Conclusions

In this study, novel artificial derivatives of organic compounds were synthesized and characterized. The resulting compounds, belonging to different organic chemical classes, included 1-O-(Trimethylsilyl)-2,3,4,6-tetra-O-acetyl-β-d-glucopyranose, 1-[(1,1-dimethylehtyl)diphenylsilyl]-1H-indole, O-tert-butyldiphenylsilyl-(3-hydroxypropyl)oleate, and 1-O-tert-Butyldiphenylsilyl-myo-inositol. These compounds could serve as mass standards for potential silicon-based plant metabolites. All the employed techniques (NALDI, FT-ICR-MS, and QqQ ESI-MS/MS) enabled the identification and characterization of each analyzed compound (**1**–**4**). Furthermore, both FT-ICR and ESI-QqQ-MS/MS facilitated a detailed examination of the fragmentation pathway of Compound **3**: O-tert-butyldiphenylsilyl-(3-hydroxypropyl)oleate. The fragmentation patterns of the remaining compounds were identified using the QqQ ESI-MS/MS system. Conversely, the application of the NALDI-MS technique revealed the presence of sodium and silver adduct ions in the investigated compounds. As a result, it can be confirmed that silicon-based silyl organic compounds can be readily identified through the use of various mass spectrometric techniques.

## Data Availability

Not applicable.

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
