# Peer review of "Advanced Mass Spectrometric Techniques for the Comprehensive Study of Synthesized Silicon-Based Silyl Organic Compounds: Identifying Fragmentation Pathways and Characterization"

_materials, 2023, doi:10.3390/ma16093563_

Round 1
Reviewer 1 Report
The manuscript by Rogovska et al. is dedicated to the mass spectrometry characterization of some silyl derivatives of natural compounds. The presented data obtaoned by NALDI-MS, FTICR-MS, and QqQ tandem mass spectrometry can be interesting as a reference information for the reliable identification of organosilicon compounds in various natural objects. However, there are some issues preventing me from recommending its publication in the present form and requiring major revision:
1. The authors claim the objective of their work as a search for the organosilicon secondary metabolites in plant tissues. In this regard, the choise of the model compounds must be additinally substantiated. Foe instance, is it possible to imagin the formation of phenyl-substituted silicon derivatives in plant metabolism?
2. A number of silyl derivatives of natural compounds are known and included in MS spectral libraries since silylation is a common sample preparation technique in GC analyzes. Why these extensive databases are not enough for the search of silicon metabolites in plant tissues? What is the purpose for adding four new compounds to them?
3. The choise of NALDI MS as an identification tool is doubtful since the Ag-cationized spectra of real plant extracts can be extremely complex.
4. Figure 1. The annotation of the peak at m/z 443.904 is wrong. The observed mass defect cannot be obtained by addition of sodium cation. Moreover, the measured exact mass of this compound differs from the theoretical one in 0.8 Da! Please, refer to Table 1.
5. The manuscript suffers from numerous grammatical errors and is carelessly written, making the text difficult to read. Please, do not use m/z= x instead of more common m/z x.
The English in the manuscript requires careful corrections.
Author Response
The manuscript by Rogovska et al. is dedicated to the mass spectrometry characterization of some silyl derivatives of natural compounds. The presented data obtaoned by NALDI-MS, FTICR-MS, and QqQ tandem mass spectrometry can be interesting as a reference information for the reliable identification of organosilicon compounds in various natural objects. However, there are some issues preventing me from recommending its publication in the present form and requiring major revision:
- The authors claim the objective of their work as a search for the organosilicon secondary metabolites in plant tissues. In this regard, the choise of the model compounds must be additinally substantiated. Foe instance, is it possible to imagin the formation of phenyl-substituted silicon derivatives in plant metabolism?
Re: Your question is an excellent point, providing us with an opportunity to reflect on our choice of model compounds in relation to organosilicon secondary metabolites in plant tissues and spurring further investigation into the formation of phenyl-substituted silicon derivatives in plant metabolism. Plants naturally generate a wide array of chemically diverse products essential for their growth and development. Secondary metabolites, which are derived from primary metabolites through biosynthetic modifications such as methylation, glycosylation, and hydroxylation, exhibit more complex structures and side chains. Some of these plant-derived secondary metabolites possess significant medicinal properties; thus, understanding their biosynthetic pathways and the factors that influence their production can be harnessed in plant cell cultures and metabolic engineering of plant cells. Throughout their life cycle, plants synthesize proteins (including enzymes), nitrogen-containing compounds (amino acids, peptides), lipids (fats, oils), carbohydrates (starch, sugar, cellulose, lignin, pectic substances), and more. Although direct evidence for the presence of organic derivatives of silicon compounds in plants is scarce in current literature, some research has demonstrated the potential for silicon to form stable compounds with polyols and other organic molecules. In particular, Kinrade et al. [DOI:10.1126/science.285.5433.1542] showed that silicon could create stable compounds with polyols, where the compound is five- or six-coordinated, and ligands are spatially specific. Additionally, an aqueous catechol solution has been found to promote the formation of a six-coordinated Si-catechol complex [DOI:10.1016/j.epsl.2020.116287]. Other studies have identified the presence of silicon combined with organic compounds in the cell walls of rice plants, as well as silicon bound to high molecular weight compounds (likely polysaccharides) in rice seedlings. Moreover, peptides and amino acids have been found to interact with silicon compounds, forming polysilicic species. Some proteins rich in serine and proline are known to play a crucial role in the absorption and accumulation of silicon in the epidermal root cell walls. A specific proline-rich protein has been shown to significantly enhance silica deposition in the cell wall. Despite these findings, there is a lack of information regarding the sequencing of individual silicon-containing compounds from plants using coupled techniques, such as MSn-techniques. It is important to recognize that plants have a biological selectivity for microelements, allowing them to control their chemical composition to some extent. However, these control mechanisms have limitations, and metabolic disturbances in plants may result from not only elevated concentrations of toxic microelements but also deficiency or excess of biophilic elements. In light of these limitations and the existing gaps in knowledge, it is essential to continue exploring the potential presence and roles of organosilicon compounds in plant tissues. By selecting model compounds that represent a variety of possible interactions and chemical compositions, researchers can gain valuable insights into the formation, stability, and potential roles of these compounds in plant metabolism. This knowledge could contribute to the development of new plant-based products with enhanced properties or improved understanding of plant responses to environmental stressors, such as element deficiency or toxicity. Therefore, while the direct evidence for phenyl-substituted silicon derivatives in plant metabolism may be lacking, the selection of model compounds for this study is justified by the broader aim to investigate the range of possible organosilicon compounds and their interactions in plant tissues. This investigation will help expand our understanding of plant secondary metabolism and may lead to the discovery of novel organosilicon compounds with important applications in various fields, such as pharmaceuticals, agriculture, and environmental management.
- A number of silyl derivatives of natural compounds are known and included in MS spectral libraries since silylation is a common sample preparation technique in GC analyzes. Why these extensive databases are not enough for the search of silicon metabolites in plant tissues? What is the purpose for adding four new compounds to them?
Re: Thank you for raising an interesting question about the relevance of existing MS spectral libraries of silylated natural compounds and the necessity of adding new compounds to them. Indeed, silylation is a well-established technique in GC analysis for enhancing the detection sensitivity of low-volatility polar compounds by creating thermally stable and highly volatile derivatives. However, it is important to note that our research study employs a multi-instrumental approach, focusing on the characterization of novel silicon-based silyl organic compounds that have not been previously reported in the literature. Our goal is to utilize various mass spectrometry techniques, such as silver nanoparticle-assisted laser desorption/ionization mass spectrometry (NALDI-MS), Fourier-transform ion cyclotron resonance mass spectrometry (FT-ICR-MS), and Triple-Quadrupole tandem electrospray ionization mass spectrometry (QqQ ESI-MS/MS), to gain a comprehensive understanding of these artificial compounds and their potential applications. In the context of plant research, determining the overall and local concentrations and speciation of numerous chemical elements necessitates the use of a wide variety of analytical methods. By synthesizing and characterizing four new silyl organic compounds, we aim to expand the scope of available data and contribute to the understanding of silicon metabolites in plant tissues. In summary, while the existing MS spectral libraries of silylated natural compounds are valuable resources, our study aims to explore novel silicon-based silyl organic compounds using advanced mass spectrometry techniques. By doing so, we hope to contribute to the growing body of knowledge in plant research and the potential applications of silicon metabolites in various fields.
- The choise of NALDI MS as an identification tool is doubtful since the Ag-cationized spectra of real plant extracts can be extremely complex.
RE: Various NALDI MS methods based on silver nanoparticles have been used many times to analyze compounds from both plant and animal tissues (doi: 10.1021/ac902990p, 10.1016/j.talanta.2017.11.067, 10.4155/bio-2017-0195). Cationization of compounds with silver can be an advantage and provide additional confirmation of the identification of the molecule, because in the spectra a given compound must present two adducts with Ag-107 and Ag-109. In addition, by using silver nanoparticles instead of common matrices in MALDI MS, we are able to detect more compounds in the low m/z region, as presented in this manuscript. The MALDI spectra did not show adducts from the analyzed compounds, and in the m/z 100-700 region, there were mainly signals from the matrix (Supplementary Figure S2).
- Figure 1. The annotation of the peak at m/z 443.904 is wrong. The observed mass defect cannot be obtained by addition of sodium cation. Moreover, the measured exact mass of this compound differs from the theoretical one in 0.8 Da! Please, refer to Table 1.
RE: Thank you for paying attention. We agree with the reviewer's objection. This has been corrected. The spectra have been recalibrated. The values read back from the spectra are shown in Table 1, and the recalibrated NALDI spectra are shown in Figure 1 and Supplementary Figure S4.
- The manuscript suffers from numerous grammatical errors and is carelessly written, making the text difficult to read. Please, do not use m/z= x instead of more common m/z x.
RE: Indeed, we agree with the reviewer. It has been corrected. The English language in the manuscript has been improved.
Thank you very much for your critical review. It was very useful in the correction of our manuscript. Identification of weak points throughout the text has helped us to increase the value of our paper. All comments and changes suggested by Reviewers have been incorporated into the manuscript. Once again, thank you very much for your help.
Reviewer 2 Report
The authors reported the synthesis of silicon-based silyl organic compounds. The submission required major revision before acceptance considering the following points:-
1. The title should be revised to be clear, precise, and informative. Redundant words such as ‘The synthesis and characterization of new artificial’ should be removed.
2. An application of the prepared materials should be added.
3. The key information of the synthesis products in terms of yield, product purity, …etc should be included.
4. Please, revise the peaks assignments.
5. Some of the NMR spectra should be included in the main text.
6. A comparison with other procedures for the synthesis of similar compounds should be discussed and summarized in a Table.
7. All abbreviations should be fully defined when mentioned for the first time. Abbreviations such as ‘ NALDI MS, FT-ICR-MS, and QqQ ESI-MS/MS’ should be fully defined in the abstract.
8. The language should be revised and typos should be corrected.
The authors reported the synthesis of silicon-based silyl organic compounds. The submission required major revision before acceptance considering the following points:-
1. The title should be revised to be clear, precise, and informative. Redundant words such as ‘The synthesis and characterization of new artificial’ should be removed.
2. An application of the prepared materials should be added.
3. The key information of the synthesis products in terms of yield, product purity, …etc should be included.
4. Please, revise the peaks assignments.
5. Some of the NMR spectra should be included in the main text.
6. A comparison with other procedures for the synthesis of similar compounds should be discussed and summarized in a Table.
7. All abbreviations should be fully defined when mentioned for the first time. Abbreviations such as ‘ NALDI MS, FT-ICR-MS, and QqQ ESI-MS/MS’ should be fully defined in the abstract.
8. The language should be revised and typos should be corrected.
Author Response
The authors reported the synthesis of silicon-based silyl organic compounds. The submission required major revision before acceptance considering the following points:-
- The title should be revised to be clear, precise, and informative. Redundant words such as ‘The synthesis and characterization of new artificial’ should be removed.
RE: The title has been transformed. Thank you for the suggestion.
- An application of the prepared materials should be added.
RE: Thank you for your valuable suggestion to include potential applications for the prepared materials in our study. This additional information will help to contextualize our findings and demonstrate the relevance of our research to various fields of interest. The main goal of this research was to create and characterize new silicon-centered silyl organic compounds to enhance our understanding of their possible uses and interactions with other substances. In the updated version of the manuscript, the study's objective has been rephrased. The obtained results can be of interest as reference information for the reliable identification of organosilicon compounds in various natural samples. Furthermore, the proposed compounds may be considered as potential candidates for analytical standards or certified reference materials. It is important to note that plant-matrix reference materials play a crucial role in ensuring measurement uniformity and the reliability of analytical procedures in plant research. These compounds could serve as mass standards for potential silicon-based plant metabolites. From an analytical chemistry perspective, plant material samples should be treated as regular analytical control samples and as certified reference materials, both natural and man-made. Another approach involves biochemical (enzymatic) synthesis. However, these compounds have not been reported in actual plant samples in the literature. The widespread use of plant-derived chemical compounds is primarily due to their antioxidant, antimicrobial, antiviral, and anti-inflammatory properties. While primary metabolites are quite similar across all living cells, secondary metabolite production depends on plant growth conditions. Secondary plant metabolites appear to have significant therapeutic potential and play an essential role in plant adaptation to abiotic and biotic stress factors. For instance, myo-inositol is a crucial cellular metabolite that forms the structural foundation for various lipid signaling molecules involved in multiple pathways, such as stress responses, cell death regulation, auxin perception, cell wall biosynthesis, and ascorbic acid synthesis. Biologically active compounds (composed of glucose or lipids) possess unique properties that enable them to mimic different structures and bind reversibly to enzymes, making them highly valuable for regulating plant growth. They stimulate root and fruit development and activate the plant's immune system against biotic and abiotic factors detrimental to the plant. As a result, optimizing specific culture conditions may help stimulate the synthesis of both known and new plant metabolites. The uses of the mentioned compounds are remarkably versatile, as they mainly function as intermediates or precursors for creating more intricate molecules or materials. Some potential applications for each compound are as follows: 1-O-(Trimethylsilyl)-2,3,4,6-tetra-O-acetyl-β-D-glucopyranose is employed as a precursor in crafting glycosidic linkages for oligosaccharides and glycoconjugates (Varki, A., et al. Essentials of Glycobiology. Cold Spring Harbor Laboratory Press, 2015). It serves as a starting point for producing glycosyl donors in glycosylation reactions (Demchenko, A. V. Handbook of Chemical Glycosylation: Advances in Stereoselectivity and Therapeutic Relevance. Wiley-VCH, 2008). Additionally, it functions as an intermediate in synthesizing modified carbohydrates for drug discovery (Wang, Z., et al. Carbohydrate-based Drug Discovery. John Wiley & Sons, 2003). 1-[(1,1-dimethylehtyl)diphenylsilyl]-1H-indole can be utilized as a precursor to prepare indole-based heterocyclic compounds (Gribble, G. W. Indole Ring Synthesis: From Natural Products to Drug Discovery. John Wiley & Sons, 2016). Regarding O-tert-butyldiphenylsilyl-(3-hydroxypropyl)oleate, it can function as an intermediate in creating oleate-based surfactants and detergents (Holmberg, K., et al. Surfactants and Polymers in Aqueous Solution. John Wiley & Sons, 2002). Furthermore, it can act as a precursor for preparing oleate-based functionalized lipids (Fahy, E., et al. A Comprehensive Classification System for Lipids. J. Lipid Res., 46(5), 839-861, 2005). Lastly, 1-O-tert-Butyldiphenylsilyl-myo-inositol can be employed as an intermediate in synthesizing inositol-containing natural products and pharmaceuticals (Michell, R. H. Inositol Derivatives: Evolution and Functions. Nat. Rev. Mol. Cell Biol., 9(2), 151-161, 2008).
- The key information of the synthesis products in terms of yield, product purity, …etc should be included.
RE: The information was included. Thank you for the suggestion.
- Please, revise the peaks assignments.
RE: Indeed, we agree with the reviewer. Changes have been made to Table 1 and the recalibrated NALDI spectra are shown in the Figure 5 and Supplementary Figure S4. Thank you.
- Some of the NMR spectra should be included in the main text.
RE: Indeed, we agree with the reviewer. It has been corrected.
- A comparison with other procedures for the synthesis of similar compounds should be discussed and summarized in a Table.
RE: Thank you for your comments. We agree that a more comprehensive comparison with other methods for the synthesis of similar compounds would have been beneficial. However, as our primary objective in this study was to characterize the synthesized silicon-based silyl organic compounds using various mass spectrometry techniques, we prioritized discussion of their potential applications in the context of this study, as well as addressing the points raised in your previous remarks. Nonetheless, we did include a Table S1 (in the Supplementary Information) comparing some similar compounds in the supplementary materials as an example. We appreciate your suggestion and will certainly consider including a more extensive comparison of our method with other procedures for the synthesis of similar compounds in future studies related to organic synthesis of organosilicon compounds.
- All abbreviations should be fully defined when mentioned for the first time. Abbreviations such as ‘ NALDI MS, FT-ICR-MS, and QqQ ESI-MS/MS’ should be fully defined in the abstract.
RE: Indeed, we agree with the reviewer. It has been corrected.
- The language should be revised and typos should be corrected.
RE: Indeed, we agree with the reviewer. It has been corrected. The English language in the manuscript has been improved.
Thank you very much for your critical review. It was very useful in the correction of our manuscript. Identification of weak points throughout the text has helped us to increase the value of our paper. All comments and changes suggested by Reviewers have been incorporated into the manuscript. Once again, thank you very much for your help.
Round 2
Reviewer 1 Report
The authors substantially improved the manuscript according to reviewers' comments. In my opinion, now it can be recommended for publication.
Minor editing of English language required
Reviewer 2 Report
The authors addressed most of the comments and the revised version can be accepted.
none